# A-B-3—Associations and dissociations of reading and arithmetic: Is domain-specific prediction outdated?

**Viktoria Jöbstl**[1]*, **Anna F. Steiner**[1,2,3], **Pia Deimann**[4], **Ursula Kastner-Koller**[4], **Karin Landerl**[1,5]

1 Institute of Psychology, University of Graz, Graz, Styria, Austria, 2 Institute of Logopedics, FH JOANNEUM, University of Applied Sciences, Graz, Styria, Austria, 3 Institute of Early Childhood and Primary Teacher Education, University College of Teacher Education Styria, Graz, Styria, Austria, 4 Department of Developmental and Educational Psychology, University of Vienna, Vienna, Austria, 5 BioTechMed-Graz, Graz, Styria, Austria

* viktoria.joebstl@uni-graz.at

**Data Availability Statement:** The final version of our data statement: Data collected for this study is available on the Open Science Framework repository (doi: 10.17605/OSF.IO/H89UE).

## Abstract

Reading and arithmetic are core domains of academic achievement with marked impact on career opportunities and socioeconomic status. While associations between reading and arithmetic are well established, evidence on underlying mechanisms is inconclusive. The main goal of this study was to reevaluate the domain-specificity of established predictors of reading and arithmetic and to enhance our understanding of the (co-)development of reading and arithmetic. In a sample of 885 German-speaking children, standard domain-specific predictors of reading and arithmetic were assessed before and/or at the onset of formal schooling. Reading and arithmetic skills were measured at the beginning and end of second grade. Latent variables were extracted for all relevant constructs: Grapheme-phoneme processing (phonological awareness, letter identification), RAN (RAN-objects, RAN-digits), number system knowledge (number identification, successor knowledge), and magnitude processing (non-symbolic and symbolic magnitude comparison), as well as the criterion measures reading and arithmetic. Four structural equation models tested distinct research questions. Grapheme-phoneme processing was a specific predictor of reading, and magnitude processing explained variance specific to arithmetic. RAN explained variance in both domains, and it explained variance in reading even after controlling for arithmetic. RAN and number system knowledge further explained variance in skills shared between reading and arithmetic. Reading and arithmetic entail domain-specific cognitive components, and they both require tight networks of visual, verbal, and semantic information, as reflected by RAN. This perspective provides a useful background to explain associations and dissociations between reading and arithmetic performance.

**Funding:** This work was supported by the Austrian Federal Ministry of Education, Science and Research (KL, PD, UK-K; www.bmbwf.gv.at/en. html) and the University of Graz (VJ; www.uni-graz.at/en/). The funders had no role in study design, data collection and analysis, decision to publish, or preparation of the manuscript.

**Competing interests:** The authors have declared that no competing interests exist.

## Introduction

Reading and arithmetic have many things in common. Both are core domains of academic achievement during the first years of schooling and mastering those skills constitutes permanent benefits in terms of better career opportunities and higher socioeconomic status [1]. Early reading development entails the acquisition of symbol-sound correspondences (i.e., letter-sound mappings in case of alphabetic writing systems) and systematic decoding procedures [2]. Over time, children apply these decoding skills as a self-teaching mechanism [3, 4] to build up an orthographic lexicon. According to the Lexical Quality Hypothesis by Perfetti and Hart [5], the quality of lexical representations—and thus the efficiency with which they can be accessed and retrieved—depends on multiple associations between orthography (a visual system), phonology (a verbal system), and semantics.

Arithmetic development typically starts with acquiring number words and Arabic digits as symbolic representations of numerosities. Young children often use counting to solve simple additions and subtractions. Over time these procedural strategies get more efficient and children build up an increasingly comprehensive number fact knowledge, which can be accessed and retrieved quickly and effortlessly. According to the Triple Code Model by Dehaene [6], three separate but efficiently integrated representational codes are developed: (1) The visual Arabic number form (e.g., 1, 2, 3), (2) the auditory verbal word frame for number words and facts (e.g., one, two, three; the time table), and (3) the analogue magnitude representation, which represents numerical quantities (e.g., five dots are more than three dots) and is the only code in the model that is thought to be semantic. The quality of numerical and arithmetic knowledge strongly depends on the efficiency and flexibility with which an individual can switch between these representations.

At this point, at least two similarities between learning to read and learning to calculate are obvious: First, both developmental domains involve a switch from early effortful procedural strategies (decoding, counting) to advanced effortless and automatized retrieval-based strategies (word recognition, number fact retrieval). And second, both skills require the build-up of integrated neurocognitive networks of verbal, visual, and semantic processing as postulated in the Lexical Quality Hypotheses of reading [5] and the Triple Code Model of numerical cognitions ([6]; Fig 1).

Evidence from brain imaging studies suggests both specific and shared reading and arithmetic networks. While reading is related to activation in the left inferior frontal gyrus and the left fusiform gyrus [7], arithmetic is based on a distributed (mostly bilateral) network of prefrontal, parietal and occipitotemporal regions (see [8] for a recent review). Importantly, functional activity in the left angular gyrus, an area shown to mediate aspects of memory retrieval (phonological representations of familiar words [9] and of arithmetic facts [10]) was found to be associated with both reading and arithmetic and is assumed to be involved in their overlap [8]. Interestingly, the few existing studies on structural and functional differences in learning disorders have so far failed to identify group differences between dyslexia and dyscalculia ([11–13]; but see [14]).

On the behavioral level, reading and arithmetic performance appear highly correlated throughout development ($r$s between .41 and .77; [15–17]), and learning disorders in these two domains (dyslexia, dyscalculia) co-occur three to five times more often than what would be expected based on individual prevalence rates [15, 18–20]. Still, the cognitive mechanisms that are driving both associations and dissociations between reading and arithmetic skills are not yet well understood.

For a long time, reading and arithmetic were mostly investigated in separate studies (e.g., [21–23]) and studies that did investigate the overlap often focused on clinical samples with

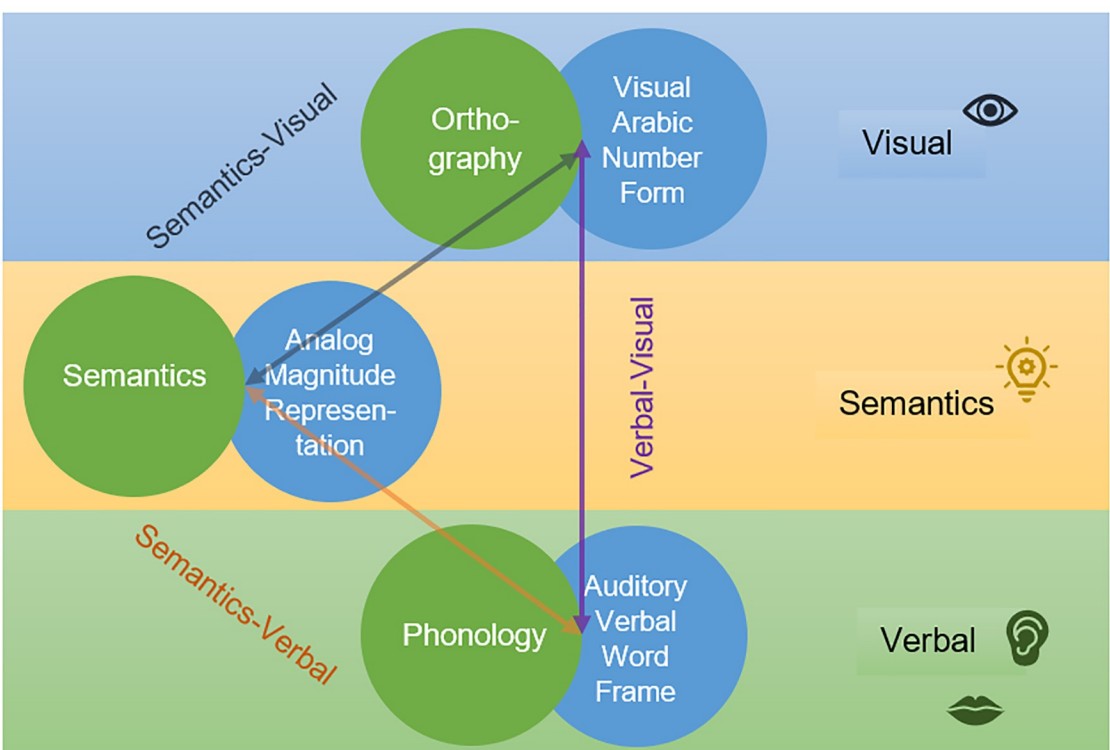

**Fig 1. The Lexical Quality Hypotheses and the Triple Code Model.** A visualization of the Lexical Quality Hypotheses of reading left [5] and the Triple Code Model of numerical cognition right [6].

dyslexia/dyscalculia (e.g., [24–26]). Recently, a new strand of research has emerged that examines cross-domain associations of common predictors of reading and arithmetic in large non-clinical samples. Unfortunately, the findings so far have been inconclusive (e.g., [24–27]). The majority of cross-domain studies investigated whether predictors are specific to one or shared between domains. However, the very same predictors may predict reading and arithmetic performance for very different reasons. In order to identify overlapping mechanisms it is crucial to investigate predictors of reading and arithmetic covariance, which was so far only addressed by a small number of studies (e.g., [24, 28, 29]). In addition, it is not yet clear why reading and arithmetic development dissociate in many young children (see [30] for a study of older children). In order to reveal the mechanisms underlying the (co-)development of academic skills, it is particularly important to focus on the early school years when foundational skills of reading and arithmetic as well as their co-occurrence are established.

Current research has identified a number of cognitive predictors of reading and arithmetic. Domain-general predictors like intelligence, working memory, or language, are well known predictors of academic attainment, including reading and arithmetic [31–33]. In the current study, we focus on predictors that are assumed to be specific to one domain of academic learning, including phonological awareness, letter knowledge, and rapid automatized naming for reading (see [21] for a recent review), and number knowledge, magnitude comparison, and counting for arithmetic (see [22] for a recent overview). These predictors have been mainly identified by single-domain studies, leading to the impression that they are "domain-specific". However, recent cross-domain studies suggest that some of these variables may be predictors

for both reading and arithmetic. The current evidence on these predictors is reviewed in the following.

## Phonological awareness

The ability to identify and manipulate phonological segments is the most prominent and consistently confirmed predictor of reading [21], whereas its contribution to arithmetic is as yet controversial. Learning to read requires precise and automatized mappings of written symbols onto sublexical phonological segments (e.g., phonemes). During early development, phoneme awareness and letter knowledge are closely interconnected [21, 34, 35], so that some studies present them as an integrated latent variable (e.g., [36, 37]). According to the Lexical Quality Hypothesis [5], efficient word recognition requires multiple and redundant associations not only between verbal phonemes and visual orthography, but also with the semantics of words. Similarly, the verbal word frame in the Triple Code Model of numerical cognition plays an important role [6, 38]: It is involved in cross-format associations between Arabic numbers and number words as well as fact knowledge about frequent additions and multiplications, which is also stored phonologically. Note, however, that the phonological segments necessary for arithmetic are typically lexical (number words), but not sublexical (syllables and sounds).

The evidence for phonological awareness as a predictor of arithmetic is mixed. Several single-domain studies predicting arithmetic with no reading measures included reported phonological awareness to be a significant predictor [39–41]. Most importantly, a longitudinal study by Hecht and colleagues [42] found phonological awareness to be predictive of arithmetic growth from $2^{nd}$ to $5^{th}$ grade. When reading skills were controlled for, phonological awareness still predicted growth from $3^{rd}$ to $4^{th}$ grade arithmetic. Cross-domain studies measuring phonological awareness and arithmetic during early developmental periods concurrently (e.g., [29, 43, 44]), or less than a year apart (e.g., [45, 46]), also observed associations. But when arithmetic was assessed several years later, no such prediction was observed [27, 44, 47], supporting the view that phonological awareness is a domain-specific predictor of reading.

In analyses of the covariance among reading and arithmetic, a similar picture has emerged: Phonological awareness explains covariance during early school development, but only when phonological awareness and arithmetic are measured at a similar time point [24, 28, 29, 45]. For instance, a study based on a large sample of 1335 Finnish children [28] assessed cognitive predictors in kindergarten and reading and arithmetic in Grades 1 and 7. In a structural equation model, the skill common to reading and arithmetic was extracted in terms of a latent covariance-variable. Phonological awareness was only predictive of covariance in $1^{st}$ grade, but not in Grade 7. The authors argued that phonological awareness might be relevant early on, when reading and arithmetic are not yet automatized (not dependent on larger units of the lexicon) and rely more on basic skills (reading: decoding, arithmetic: verbal counting). Phonological awareness measured at the same time point early in development seems to partly capture some variance of overlapping phonological processes involved in reading as well as arithmetic (e.g., [29, 43, 44]), but it is unlikely that it is directly linked to arithmetic development, given the lack of studies documenting longitudinal associations [27, 44, 47].

## Letter knowledge

To learn letters and Arabic numbers, a child needs to establish arbitrary mappings between specific visual symbols (e.g., M and 1, respectively) and the corresponding verbal information ("m" and "one"). In order to use letters productively for reading, children need to understand that they represent specific phonological units in their spoken language. This explains why letter knowledge is closely tied to phonological awareness, especially during early development

[21]. Early letter knowledge was repeatedly found to predict reading and arithmetic, as well as their covariance [24, 26–28]. The association of letter knowledge with arithmetic is often explained by the fact that arithmetic also requires efficient symbol knowledge in terms of Arabic digits (see below), but number knowledge is rarely assessed in these studies (see [29, 44, 48] for studies assessing both).

## Rapid automatized naming and counting as indicators of serial retrieval fluency

Proficient reading and arithmetic rely on fast and accurate lexical retrieval. For reading, this requires a shift from decoding based letter-sound correspondences (i.e., sounding-out) to retrieving the pronunciation of whole written words from memory (i.e., lexical reading). For arithmetic, it shifts from arithmetic procedures like (finger) counting to the retrieval of arithmetic facts, which increases arithmetic fluency (e.g., remembering "2 + 3 = 5" instead of calculating "2 + 3"). Rapid automatized naming (RAN) seems to mirror processes of serial retrieval of visual-verbal associations particularly well [49]. In this paradigm, sets of visual stimuli (e.g., letters, digits, objects, colors) are named sequentially as quickly and accurately as possible. The relationship between RAN and reading has been studied extensively, but the components driving this associations are not yet clear. Various theories have been proposed, including processing speed, phonological, and orthographic processing, but even after controlling for these factors, RAN remained a significant predictor of reading (for a review [50]). Serial and verbal components of RAN-tasks seem to play a role ([49, 51]) but the exact nature of RAN measures is still unclear.

RAN has long mostly been discussed as a domain-specific predictor of reading [21]. More recently, several cross-domain studies also linked RAN to arithmetic fluency (e.g., [46, 47, 52, 53]) as well as to the covariance of reading and arithmetic (e.g., [24, 28, 45, 46]). Findings of studies using untimed arithmetic tasks are less straightforward (significant: [25, 54]; indirect effect: [26], non-significant: [48, 55]).

A Finnish study [45] observed a high correlation (.67) of 1st graders' RAN performance with a measure of verbal counting fluency, and combined the two into a latent variable of Serial Retrieval Fluency, as both skills depend on fast and efficient retrieval of sequential information from long term memory. Indeed, Serial Retrieval Fluency predicted the shared fluency in reading and arithmetic in Grade 2. Other studies also found associations of verbal counting with reading (e.g., [30, 47, 56]) and the covariance of reading and arithmetic (e.g., [28, 29, 45, 48]). The relation is perhaps plausibly explained by verbal processing and serial retrieval skills that play a role in both verbal counting and reading.

## Number knowledge

The development of the number concept usually starts with learning small number words and quantitative meaning at the age of two or three years [57, 58]. By the age of four, children begin to master the associations between quantities, digits, and number words (e.g., [59]). In addition, children need to understand the complexities of multi-digit numbers and place-value representation [60, 61].

Number and letter knowledge are moderately to strongly correlated ($r$ = .38-.62; see [29, 44, 45, 48] for details). The assumption that number knowledge depends on similar processes as letter knowledge is confirmed by the fact that both explain variance in reading [26–28] as well as arithmetic [24, 44], and both predict variance shared between these domains of academic learning (letter knowledge: [28, 29]; number knowledge: [45]). However, in studies that assess both number and letter knowledge, the two predictors might account for the same variance

and the stronger predictor might nullify the weaker one. Indeed, one study observed a prediction of letter but not number knowledge [29], while another study reported the reverse pattern [44]. Another issue that could have an impact on predictive patterns is that assessments of number knowledge vary greatly between studies (naming vs. reading of numbers, single vs. multi-digit and length of multi-digit numbers).

## Magnitude comparison

Within the Triple Code Model [6], the analog magnitude representation provides the "number sense", i.e., the semantics of numbers, in which the two symbolic types of representations (Arabic numbers and number words) are tightly connected. Magnitude comparison assesses how efficiently children process numerical magnitude. In such paradigms children select the numerically larger of two sets of dots (non-symbolic condition) or Arabic digits (symbolic condition). While processing magnitudes is obviously important for arithmetic, and there is ample evidence that magnitude comparison skills are related to arithmetic performance [62], reasons for an association with reading are less apparent. Still, in a sample of 193 English-speaking children [24], symbolic magnitude comparison in kindergarten explained variance in 1$^{st}$ grade arithmetic as well as reading comprehension, even when counting, number knowledge, phonological awareness, and RAN were taken into account. However, the effect was small ($\beta = .13$, $p = .044$), and in the same study symbolic digit comparison did not explain unique variance in decoding or reading fluency. Other studies using non-symbolic (e.g., [63]) or symbolic comparison (e.g., [64]) as predictors did not find a significant association with reading. Regarding skills shared between reading and arithmetic, Koponen et al. [45] found that symbolic magnitude comparison explained unique variance in reading and arithmetic covariance, even when other serial retrieval tasks were considered (RAN, verbal counting). Note that digit comparison requires fast retrieval of alphanumeric symbols, which may involve similar processes as retrieving words during reading. As magnitude comparison is rarely investigated in research on reading development, more evidence is needed to discard or support these isolated findings.

## Current study

The aim of the current study was to investigate which predictors that have been considered to be domain-specific in earlier research are indeed specific to either reading or arithmetic, and which predictors are associated with both. Even though many questions remain unanswered as yet, the studies outlined above have provided robust evidence to support cross-domain associations. This is to be expected given marked reading-arithmetic correlations between .41 and .77 [15–17]. However, these correlations also indicate that there is between 43 and 83% variance that is not shared between the two domains. It is crucial to better understand the patterns of cognitive prediction, especially during early periods of academic learning, when the foundations of the association are established.

We assessed predictors twice before formal teaching of reading and arithmetic starts (about eight months before and again right at the onset of formal schooling). Such an early assessment can reduce confounding effects of interdependences between skills, such as the reciprocal relation between phonological awareness and reading (e.g., [21, 65]). It also allows early identification of potential strengths and weaknesses facilitating individualized instruction and timely intervention where necessary.

In order to reduce the influence of measurement error and effects due to specific task characteristics, latent variables for all relevant constructs were extracted from both assessments and two different tasks (see Materials & methods for details). Four predictor variables were

introduced: (1) grapheme-phoneme processing (indicated by phonological awareness and letter identification), as a measure of children's ability to process sublexical sound units and their mapping with specific visual symbols (i.e., letters). This component thus relates to the association between phonology and orthography in Fig 1; (2) RAN (objects and digits), indicating visual-verbal associations and their serial retrieval is an important component in both reading as well as arithmetic (see Fig 1); (3) number systems knowledge (number identification and successor knowledge) as an indicator of children´s concepts of the number word system (i.e., the verbal word frame in Fig 1) and how it is represented by single- and multi-digit Arabic numbers (i.e., the visual-Arabic code in Fig 1), and (4) magnitude processing (non-symbolic and symbolic), indicating mandatory processing of analog magnitudes.

Reading and arithmetic performance were assessed at the beginning and end of Grade 2, and latent variables for each construct as well as a covariance variable were again extracted across the two assessment points (see [47, 56] for a similar approach). We applied three different approaches to investigate the predictors of associations and dissociations between reading and arithmetic, addressing different research questions:

1. In a general approach (Model 1), reading and arithmetic fluency were introduced as separate dependent (latent) variables, to explore which cognitive predictors are linked to each domain. This model thus tests the assumption that certain predictors of reading and arithmetic are domain-specific. Given the structure of Model 1, results will be comparable to studies investigating only one domain of achievement. We expected grapheme-phoneme processing to predict reading. We also expected number system knowledge and magnitude processing to predict arithmetic. Based on earlier findings, we expected RAN to be related to both domains of academic learning.

2. In two further models we tested which predictors explain skills unique to reading or arithmetic by including the other academic achievement domain as an additional predictor. In Model 2, reading was the dependent variable and arithmetic was added as a predictor. We expected grapheme-phoneme processing to contribute to variance in reading because access to sublexical phonological units is necessary for reading alphabets but not for doing arithmetic. We also expected RAN to contribute to variance in reading because serial retrieval, which is relevant for both academic skills, also shares a naming component with reading (aloud). In Model 3, arithmetic was the dependent variable and reading was an additional predictor. We expected number knowledge and magnitude processing to show specific associations with arithmetic performance once all variance shared with reading is accounted for.

3. The third approach (Model 4) is modelled after studies that have used reading and arithmetic as indicators of a latent covariance-variable (e.g., [28, 29, 45, 48]). This approach identifies predictors that explain shared difficulties or strengths in reading and arithmetic. In line with these studies, we expected RAN to be a predictor of covariance, as efficient visual-verbal retrieval is required for both academic skills. We also expected that grapheme-phoneme processing and number system knowledge would be predictors of covariance since both involve knowledge of mappings between visual symbols and verbal information. Magnitude processing was not expected to predict covariance because it is specific to the component of analog magnitude representation, even though the symbolic condition of this paradigm involves processing of arbitrary symbols.

Integrating the pattern of findings across these three distinct approaches will provide us with a detailed picture of the predictive mechanisms underlying reading, arithmetic, and their association.

## Materials and methods

### Participants

Children from 25 schools across five Austrian states were followed from their final kindergarten year to 2nd grade. In spring before entering 1st grade (t1; 2019), we assessed established cognitive predictors of reading and arithmetic in 638 children ($M_{age}$: 5 years 11 months; $SD_{age}$: 3.59 months; 50% females; 56% monolingual speakers of the instructional language German). About eight to nine months later, right at the beginning of 1st grade (t2), the same assessment was rerun with 570 children of the initial sample ($M_{age}$: 6 years 7 months; $SD_{age}$: 3,71 months; 51% females; 57% monolingual German). At this point (t2), we were able to recruit 247 additional children ($M_{age}$: 6 years 9 months; $SD_{age}$: 5.57 months; 49% females; 47% monolingual German) from the same schools, who completed the task battery for the first time. No systematic differences were observed between children entering the study at t1 or at t2 on any of the available assessments (group comparisons for all variables using alpha-adjusted t-tests).

Reading and arithmetic performance were assessed at the beginning of 2nd grade (t3, $n = 600$), and again at the end of the same school year (t4, $n = 673$). The main reasons for missing data in 2nd grade were school closures due to the corona pandemic ($n_{t3} = 76$), being enrolled in an additional preschool year at the end of the kindergarten period ($n = 87$) or moving to a different school ($n_{t3|t4} = 68|77$). Written consent was obtained from a legal guardian and the project was approved by the ethics committee of the University of Graz as well as local and national school authorities.

**Austrian school system and pandemic-related school closures.** Children in Austria enter the primary school system in September after their sixth birthday. The Austrian kindergarten system is independent from the school system, and only the last kindergarten year is mandatory. Kindergarten activities vary greatly, and mostly focus on social and language skills, while activities including letters or numbers are highly exceptional. Formal instruction in reading and arithmetic starts in Grade 1. About eight months before they start school, all children attend a mandatory school enrolment session in their future primary school. The aim of this session is to identify children with marked developmental delays who are then enrolled in an additional preschool year in their primary school, before they move on to 1st grade. The first assessment in the reported longitudinal study (t1) was carried out during this school enrolment session.

Note that the school cohort followed in the current study was affected by two periods of pandemic-related school closures: Children received distance teaching at home towards the end of Grade 1 (March–May 2020) and again in the middle of Grade 2 (November–December 2020). We acknowledge that school achievement may have been lower than in other cohorts [66, 67] and parental support may have had a stronger influence [67, 68]. However, we have no reason to assume that the association between reading and arithmetic or the patterns of cognitive prediction should have been affected.

### Procedure

This longitudinal study was part of a larger project aimed to develop and validate a screening tool to identify children at risk for learning problems early on. Only variables relevant for the current research questions are described in the method section. All assessments were carried out in children´s primary schools.

Assessments of cognitive predictors (t1 and t2) were carried out by trained teachers and psychology (under)graduate students (overall duration of the assessment was about 30–40 minutes). All tasks were embedded in a story line where a gnome named "Poldi" asked

children to help her find a hidden treasure. After each task, irrespective of performance, the child received a symbolic key which opened a treasure chest at the end of their journey. All tasks were available in paper-pencil and digital format (via tablet), with the exception of phonological awareness, RAN (objects and digits), digit comparison, and successor knowledge at t1, which were only available as paper-pencil tasks. It was generally the schools that decided whether they wanted to use the digital or the paper-pencil version. If a school decided to use the digital version at t1, five tasks were completed via paper-pencil format and the remaining via tablet (45%). At t2, a child either completed all tasks via tablet (70%) or paper-pencil format. In the digital version gnome Poldi and not the examiner introduced the tasks. Some cognitive tasks had parallel versions (A & B), which differed in item order only (phonological awareness, symbolic magnitude comparison). For other tasks the parallel versions A and B also had different item sets (letter identification, number identification, successor knowledge, t1 non-symbolic magnitude comparison). Again, each child either completed version A (t1: 48%; t2: 53%) or version B. With one exception (t1 letter identification—see task section for details) mean item difficulties were similar and scores were comparable between parallel versions and presentation formats (multivariate analysis of variance with "version" or "format" as fixed factor—see [69] for details).

The assessments of reading and arithmetic at the beginning and end of Grade 2 (t3 and t4) were administered by trained (under)graduate psychology students and psychologists. Word and pseudoword reading were administered individually (~ 15 minutes), while sentence reading, one-minute addition, subtraction, and the numerical operations task were carried out in class (~ 60 minutes).

## Tasks

**Phonological awareness.**   In an onset identification paradigm, children were instructed to identify a word with a certain onset phoneme from three alternatives. First, a word and its initial phoneme were verbally presented (e.g., "Affe beginnt mit /a/"; Engl.: "Monkey starts with /. . ./."). Then three pictures of familiar objects were presented (to keep memory load to a minimum) and named (e.g., "Welches Wort beginnt noch mit /a/: Ast, Baum, Mast."; Engl.: Which word also starts with /. . ./: branch, tree, mast."). Children gave their response by pointing at/ tapping on the corresponding picture. No feedback was given. The target word and one of the two distractors always rhymed (e.g., Ast—Mast), while the other distractor had no similarity. The total score was the number of correct responses for the 12 items. Cronbach's alpha was .78 for t1 and .76 for t2.

**Letter identification.**   The task was to identify a specified letter among four to six alternatives presented side by side in one row. The phoneme corresponding to the target letter was verbally presented (e.g., "Where is /u/?") and children pointed to the corresponding letter. Letters were presented in upper or lower case. For the first two items, distractors were letters and digits (e.g., /u/ → 5, V, 8, U). Four further items had only letters to choose from and the final two items were syllables (e.g., /sa/ → KA, SA, SO, MA, AS). One t1 item showed a clear ceiling effect. To keep results comparable between parallel versions, the outlier item was excluded and the percent of correct responses was computed for the two parallel versions at t1. The score at t2 was the number of correct responses across all items (max. = 9). Cronbach's alpha for parallel version A was .77 at t1 and .71 at t2. Cronbach´s alpha for parallel version B was .80 at t1 and .71 at t2.

**Rapid automatized naming (RAN).**   Rapid automatized naming was assessed in two conditions: RAN-objects and RAN-digits. During a practice phase, five pictured objects or digits were presented in one row and the examiner ensured that the child could name all items

correctly. Ninety-eight children were not yet able to name all digits at t1, so this condition was discontinued. On a separate page, the same set of five items was repeated seven times, pseudo-randomly arranged in each of seven rows. The child was instructed to name the items as fast and accurately as possible. The time needed to name all 35 items (t1) or the number of items named within a time limit of 20 seconds (t2) was recorded. Errors were exceptional (< 1 error per condition on average) and were therefore not considered for further analyses. The score was the number of items per second. Retest reliability across the approximately eight months between t1 and t2 was .61 for RAN-objects and .67 for RAN-digits.

**Number identification.**   The number identification task was adapted from Göbel et al. [70] and was analogous to letter identification. The number word was verbally presented (e.g., "Where is four?") and the child was asked to select the corresponding Arabic number from four to six response alternatives. At t1, three single-digit, four two-digit and two three-digit numbers were presented ($n = 9$). At t2, there was one two-digit-number less ($n = 8$). Distractors were chosen based on visual similarity and common difficulties regarding place value notation (see [61, 71]). The score was the number of correct responses. Cronbach's alpha was .71 at t1 and .51 at t2 for parallel version A and .73 at t1 and .55 at t2 for version B.

**Successor knowledge.**   In four items, children were asked to name the preceding or following number in a word problem format (e.g., "My sister is five years old. How old will she be next year?"). The items included one (version A) or two (version B) one-digit numbers and three (version A) or two (version B) two-digit numbers. Version A was composed of one predecessor (n– 1; one-digit item) and three successor items (n + 1), while version B was composed of two predecessors (one-digit item) and two successor items. The score was the number of correct responses (max. = 4). Cronbach´s alpha was .72 at t1 and .74 at t2 for parallel version A and .78 at t1 and .72 at t2 for version B.

**Magnitude comparison.**   Non-symbolic and symbolic magnitude comparison were assessed in two separate conditions. The basic requirement was always to select the numerically larger of two sets of dots (non-symbolic condition) or digits (symbolic condition). For both conditions, task variants were slightly changed throughout the study period in order to accommodate time restrictions in schools and develop short and efficient measures. In the non-symbolic condition children selected the larger among two sets of squares, presented as coins. Each set consisted either of between 7 and 11 same-sized squares with a numerical distance of one or two (ratio: 6:7) between the two sets or between 20 and 40 squares with a numerical distance of six to ten (ratio: 6:10) covering the same surface in both displays (with different sizes of individual squares). The latter item type turned out to be quite difficult and was no longer used at t2. In the digital version, two sets of squares (presented as coins) were displayed side by side and the child was instructed to tap on the larger set as quickly as possible, without counting. For the paper-version, a DIN A4 booklet (21 cm x 29.7 cm) with eight dot pairs per page underneath each other was presented and children pointed to the larger set as quickly as possible, without counting. At t1, 22 dot pairs were given and the response times summed across all comparisons was recorded. At t2, a time limit of 30 seconds was introduced (paper-version) and children responded to as many of overall 24 items as possible. In the digital version only the first 16 out of 24 pairs were presented and again, the response time was recorded and summed across items.

In the symbolic condition, children were asked to select the numerically larger of two single digits. At t1 (paper version only), 18 digit pairs were presented in booklet format (nine pairs per page) and the time needed to respond to all items was recorded. At t2 the digital version consisted of 18 digit pairs presented sequentially. For the paper version the number of items that were responded to within a time limit of 20 seconds was recorded.

The score for both conditions was the number of correct items minus the number of errors per second, so that fast but inaccurate children automatically obtained low scores. Because of the different task versions, raw scores were z-transformed separately for each version. The average Cronbach's alpha across task versions for the non-symbolic magnitude comparison condition was .78 at t1 and .68 at t2, and for the symbolic condition .93 at t1 and 81 at t2.

**Reading.** Reading skills at the beginning and the end of Grade 2 were assessed by two standardized tests: In a classroom sentence reading test (t3: SLS 1–4 [72]; t4: SLS 2–9 [73]), children silently read semantically and syntactically simple sentences (e.g., Elephants are heavy.). Each sentence was followed by a checkmark and a cross. For semantically correct sentences the checkmark was to be circled, for inappropriate sentences the cross. Response accuracy was very high, indicating that children had no problems to comprehend the semantic content. The main criterion was how many sentences could be responded to within a time limit of three minutes. The score was the number of correct minus incorrect responses, to control for guessing.

The second test was an individually administered one-minute word and nonword reading test (an updated version of SLRT II [74]; with minor changes to items). In two separate conditions, children were asked to read aloud as many words or nonwords in one minute as possible and the score was the number of correctly read items. Parallel-test reliabilities for this age group according to the manuals are .95 for the SLS 2–9, .92 for the SLS 1–4, .98 for word reading, and .96 for pseudoword reading.

**Arithmetic.** Three classroom measures were applied to assess 2nd grade arithmetic skills (t3 and t4). In two speeded conditions, children were instructed to solve as many simple additions and subtractions (all numbers below 20) as possible within a time limit of one minute. The third condition were written arithmetic problems adapted from the Numerical Operations subtest of the WIAT-II [75]. At t3, 12 additions, subtractions and multiplications with increasing complexity (including carry operations and multidigit numbers) were presented, at t4 the number of items was increased to 20 and two divisions were included. Children had 15 minutes to work on these problems. The score for each task was the number of correct items. The internal consistency for numerical operations calculated on the basis of this sample was .77 (t3 & t4). The retest-reliability of one-minute addition and one-minute subtraction, with a time interval of over 6 months (beginning to the end of 2nd grade) was .55 and .67, respectively.

## Analysis plan

We intended to run four separate structural equation models to investigate shared and unique predictors of reading and arithmetic. Since structural equation modelling is a confirmatory rather than exploratory approach, we first estimated parsimonious models (Models 1–4), which we specified according to our predictions. Given the young age of the children in our study, we wanted to create latent variables to account for measurement error and capture constructs as accurately as possible. Since predictors were only moderately correlated and showed moderate to low reliability (as is often the case in this age group), we used data from both time points to increase the number of indicators for each latent variable (see [47, 56] for studies using a similar approach). Thus, each of the four predictor variables was extracted from four indicators across t1 and t2 (see Fig 2): Grapheme-phoneme processing is based on performance in phonological awareness and letter identification, RAN is based on digit and object naming conditions, number systems knowledge includes number identification and successor knowledge and magnitude processing is based on the non-symbolic and symbolic comparison conditions.

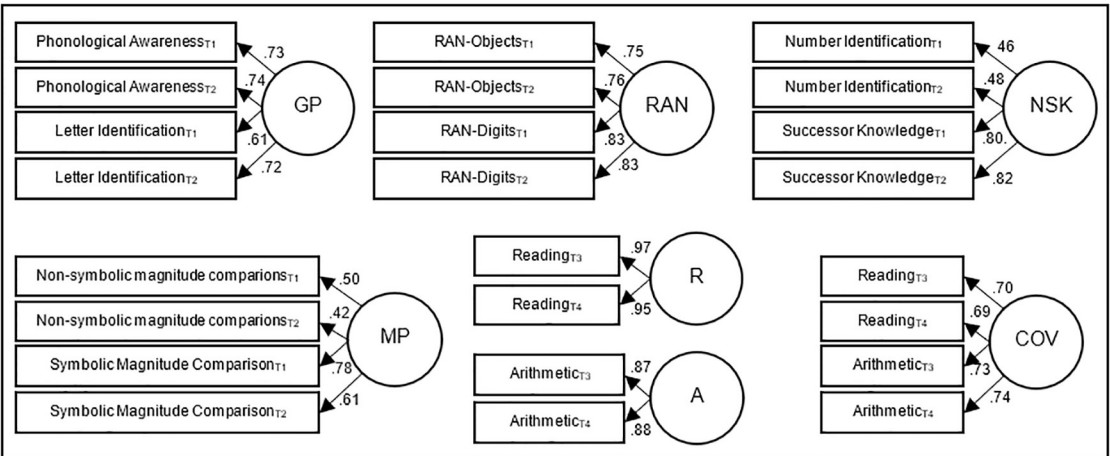

**Fig 2. Overview of all latent variables including their respective indicators.** GP = Grapheme-phoneme processing; RAN = Rapid automatized naming; NSK = Number system knowledge; MP = Magnitude processing; R = Reading; A = Arithmetic; COV = Covariance; factor loadings slightly vary between models (± .02).

The criterion measures reading and arithmetic were assessed with three different tasks per time point (t3/t4). A model using the three tasks as indicators for latent variables resulted in non-positive definite covariance matrices. At first, we assumed that this was because the assumption of local independence was not met, i.e., because of closer format overlap of two out of the three tasks (word and nonword reading had the same format of reading aloud for one minute whereas sentence reading was silent and children had to mark semantic correctness; addition and subtraction fluency had the same task format, whereas numerical operations consisted of more complex arithmetic problems with less focus on speed). We attempted to address this issue by averaging z-transformed scores of the two similar tasks (word and nonword reading; addition and subtraction fluency) and using this mean and the score of the third task as indicators of a latent variable. However, this again resulted in a non-positive definite covariance matrix, indicating that the observed variables were not related to each other in a way that could be explained by the hypothesized latent variable structure. As both reading and arithmetic tasks showed reasonable to high stability across the two assessments (reading: .849-.888; arithmetic: .550-.674), we chose to reduce the complexity of the model and use a similar approach as for the predictors by combining the two time points (t3/t4): We z-transformed all tasks separately and averaged the three reading and the three arithmetic tasks. The mean reading and arithmetic scores of the two time points were then used as indicators of latent variables for reading and arithmetic.

Importantly, we also extracted a "covariance" variable representing the skills shared between reading and arithmetic, which was indicated by reading and arithmetic at the beginning and end of 2nd grade. Fig 2 summarizes the latent variables and their indicators.

Four structural equation models were run to address the following questions: 1) Which predictor variables are associated with reading and/or arithmetic (Model 1)? 2) Which predictors explain unshared variance, i.e., variance specific to reading (Model 2) or arithmetic (Model 3)? 3) Which predictors explain skills shared between reading and arithmetic, and thereby drive association (Model 4)?

All models were run using the statistic software R (version 4.2.0; [76]) with the packages *lavaan* [77] and *semTools* [78] for structural equation modelling. The *sem.mi* function was

**Table 1. Descriptive statistics of predictor and criterion variables.**

| Variables | T1 | | | | | T2 | | | | |
|---|---|---|---|---|---|---|---|---|---|---|
| | $M$ | $SD$ | Min | Max | $N$ | $M$ | $SD$ | Min | Max | $n$ |
| Phonological awareness[a] | 7.90 | 2.98 | 0 | 12 | 630 | 8.83 | 2.75 | 0 | 12 | 798 |
| Letter identification | 44[b] | 28 | 0 | 1 | 630 | 5.60[a] | 2.22 | 0 | 9 | 800 |
| RAN-objects[c] | 0.75 | 0.20 | 0.19 | 1.41 | 604 | 0.94 | 0.23 | 0.40 | 2.06 | 762 |
| RAN-digits[c] | 0.80 | 0.27 | 0.19 | 2.47 | 510 | 1.03 | 0.30 | 0.40 | 2.33 | 710 |
| Number identification[a] | 3.90 | 1.82 | 0 | 9 | 630 | 4.26 | 1.37 | 0 | 8 | 800 |
| Successor knowledge[a] | 1.62 | 1.43 | 0 | 4 | 630 | 2.25 | 1.42 | 0 | 4 | 782 |
| Non-symbolic magnitude comparison | 0.00[d] | 1.00 | -3.87 | 4.90 | 570 | 0.00[d] | 1.00 | -4.67 | 4.19 | 793 |
| | 61[b] | 24 | 0 | 100 | 636 | 78[b] | 16 | 25 | 100 | 795 |
| Symbolic magnitude comparison | 0.00[d] | 1.00 | -3.22 | 4.49 | 536 | 0.00[d] | 1.00 | -4.78 | 2.33 | 792 |
| | 74[b] | 29 | 0 | 100 | 595 | 83[b] | 17 | 0 | 100 | 795 |
| | T3 | | | | | T4 | | | | |
| | $M$ | $SD$ | Min | Max | $N$ | $M$ | $SD$ | Min | Max | $n$ |
| Reading[d] | 0.00 | 0.94 | -2.32 | 4.41 | 597 | -0.01 | 0.93 | -2.97 | 4.26 | 673 |
| Words[a] | 33.37 | 17.32 | 0 | 119 | 589 | 48.53 | 19.15 | 0 | 132 | 671 |
| Pseudowords[a] | 25.82 | 9.41 | 0 | 70 | 589 | 32.45 | 10.30 | 0 | 75 | 671 |
| Sentences[a] | 17.99 | 11.08 | -3 | 61 | 591 | 30.32 | 11.57 | -7 | 80 | 672 |
| Arithmetic[d] | 0.00 | 0.85 | -2.24 | 2.62 | 595 | 0.00 | 0.85 | -3.07 | 3.43 | 673 |
| Additions[a] | 15.98 | 5.28 | 0 | 30 | 595 | 18.34 | 5.62 | 1 | 42 | 673 |
| Subtractions[a] | 13.66 | 5.15 | 0 | 30 | 594 | 17.91 | 5.87 | 0 | 45 | 673 |
| Operations[a] | 7.00 | 2.24 | 1 | 12 | 593 | 13.01 | 3.03 | 0 | 20 | 673 |

[a] Number of correct responses

[b] Percent correct

[c] Items/second

[d] z-score.

The differences in "$n$" between z-scores and percent correct for magnitude comparison tasks are due to missing reaction time data, while number of correct items was recorded.

used to handle missing data. The number of complete cases per variable is displayed in Table 1. We imputed five values per missing data point using the remaining variables as indicators (Bayesian linear regression). Five structural equation models were then performed, after which estimates across the imputed models were pooled for a final model. A set of five indicators was used to determine the goodness of fit (see [79–81], for statistical justification): Comparative fit index (CFI; > .95), Tucker Lewis index (TLI; > .90), root mean square error of approximations (RMSEA; < .06), standardized root mean square residual (SRMR; < .08), and the likelihood ratio test statistic ("$X^2$"). Because the likelihood ratio test statistic is sensitive to sample size, it may reject an adequate model in larger samples [82, 83]. In this study we therefore used a proposed alternative [17, 82]: The chi-square to degrees of freedom ratio ($X^2/df < 3$). Misspecifications indicated by modification indices were only applied, if they were theoretically plausible.

## Results

Descriptive statistics of all predictors and outcome variables are displayed in Table 1 and the correlation matrix in Table 2.

**Table 2. Bivariate correlation matrix of manifest variables indicating latent variables.**

| | 1 | 2 | 3 | 4 | 5 | 6 | 7 | 8 | 9 | 10 | 11 | 12 | 13 | 14 | 15 | 16 | 17 | 18 | 19 | 20 |
|---|---|---|---|---|---|---|---|---|---|---|---|---|---|---|---|---|---|---|---|---|
| **T1** | | | | | | | | | | | | | | | | | | | | |
| 1. Phonological awareness | | 630 | 604 | 510 | 630 | 630 | 570 | 536 | 555 | 557 | 531 | 486 | 557 | 552 | 554 | 554 | 425 | 421 | 469 | 469 |
| 2. Letter identification | .461 | | 604 | 510 | 630 | 630 | 570 | 536 | 555 | 557 | 531 | 486 | 557 | 552 | 554 | 554 | 425 | 421 | 469 | 469 |
| 3. RAN-objects | .394 | .288 | | 504 | 604 | 604 | 555 | 525 | 537 | 539 | 522 | 478 | 539 | 534 | 535 | 536 | 419 | 417 | 460 | 460 |
| 4. RAN-digits | .325 | .315 | .617 | | 510 | 510 | 473 | 464 | 459 | 461 | 444 | 435 | 461 | 456 | 458 | 459 | 378 | 376 | 419 | 419 |
| 5. Number identification | .329 | .553 | .219 | .211 | | 630 | 570 | 536 | 555 | 557 | 531 | 486 | 557 | 552 | 554 | 554 | 425 | 421 | 469 | 469 |
| 6. Successor knowledge | .408 | .396 | .450 | .337 | .381 | | 570 | 536 | 558 | 560 | 534 | 488 | 560 | 555 | 557 | 557 | 428 | 424 | 473 | 473 |
| 7. Non-symbolic magnitude comparison | .339 | .286 | .381 | .262 | .244 | .326 | | 499 | 501 | 503 | 482 | 443 | 503 | 500 | 501 | 503 | 383 | 381 | 425 | 425 |
| 8. Symbolic magnitude comparison | .434 | .326 | .483 | .467 | .348 | .534 | .377 | | 486 | 487 | 469 | 440 | 487 | 485 | 484 | 486 | 383 | 380 | 428 | 428 |
| **T2** | | | | | | | | | | | | | | | | | | | | |
| 9. Phonological awareness | .562 | .401 | .352 | .310 | .262 | .415 | .311 | .355 | | 797 | 758 | 707 | 797 | 779 | 790 | 789 | 577 | 576 | 643 | 643 |
| 10. Letter identification | .489 | .492 | .302 | .377 | .258 | .337 | .257 | .377 | .538 | | 760 | 708 | 800 | 780 | 792 | 791 | 579 | 578 | 645 | 645 |
| 11. RAN-objects | .321 | .245 | .606 | .519 | .188 | .347 | .247 | .333 | .368 | .341 | | 686 | 760 | 746 | 755 | 754 | 558 | 557 | 620 | 620 |
| 12. RAN-digits | .342 | .262 | .508 | .673 | .215 | .322 | .195 | .371 | .351 | .457 | .651 | | 708 | 695 | 702 | 704 | 539 | 539 | 597 | 597 |
| 13. Number identification | .261 | .168 | .249 | .316 | .262 | .318 | .106* | .357 | .304 | .353 | .220 | .338 | | 780 | 792 | 791 | 579 | 578 | 645 | 645 |
| 14. Successor knowledge | .425 | .353 | .455 | .336 | .347 | .679 | .287 | .494 | .491 | .406 | .347 | .348 | .359 | | 776 | 775 | 569 | 568 | 635 | 635 |
| 15. Non-symbolic magnitude comparison | .224 | .140 | .288 | .204 | .151 | .259 | .236 | .314 | .256 | .192 | .263 | .151 | .147 | .290 | | 790 | 578 | 576 | 644 | 644 |
| 16. Symbolic magnitude comparison | .249 | .230 | .340 | .341 | .309 | .371 | .204 | .460 | .308 | .348 | .330 | .431 | .443 | .441 | .275 | | 576 | 575 | 642 | 624 |
| **T3** | | | | | | | | | | | | | | | | | | | | |
| 17. Reading | .278 | .242 | .314 | .422 | .175 | .294 | .192 | .336 | .360 | .300 | .409 | .428 | .285 | .275 | .158 | .260 | | 594 | 580 | 580 |
| 18. Arithmetic | .153** | .172 | .269 | .421 | .161 | .304 | .219 | .377 | .272 | .212 | .331 | .384 | .284 | .264 | .216 | .368 | .504 | | 579 | 579 |
| **T4** | | | | | | | | | | | | | | | | | | | | |
| 19. Reading | .254 | .234 | .346 | .425 | .165 | .313 | .131** | .322 | .314 | .286 | .425 | .455 | .270 | .274 | .149 | .261 | .920 | .468 | | 673 |
| 20. Arithmetic | .173 | .182 | .298 | .362 | .161 | .329 | .203 | .361 | .264 | .221 | .305 | .372 | .248 | .331 | .184 | .322 | .532 | .754 | .502 | |

Above the diagonal = *n*; below the diagonal = *r*; not marked *rs* = $p < .001$;

** $p < .01$;

* $p < .05$

## Model 1—What predicts reading and arithmetic

The first model tested the general assumption that predictors of reading and arithmetic are domain-specific. We entered grapheme-phoneme processing as a domain-specific predictor of later reading and number system knowledge as well as magnitude comparison as domain-specific predictors of later arithmetic. Only RAN was assumed to predict both academic skills (Fig 3; parsimonious model). The initial fit indices indicated poor fit, $X^2(158) = 583.063$, $p < .001$, $X^2/df = 3.69$; CFI = 0.917; TLI = 0.900; RMSEA = .055 (90% CI = .050–.060); SRMR = .048. The modification indices and residual correlations made apparent that there were correlations between predictor variables that could not be reproduced by the proposed model. We therefore entered links between the tasks that relied on some number identification skill (number identification, symbolic magnitude comparison, and RAN-digits, separate for t1 and t2). Because letter identification and number identification were assessed using the same task format, we included a link between these error variances as well. This led to an increase in model fit, $X^2_{difference}(8) = 185.733$, $p < .001$ and a better-fitting model, $X^2(150) = 397.330$, $p < .001$, $X^2/df = 2.65$; CFI = 0.952; TLI = 0.939; RMSEA = .043 (90% CI = .038–.048); SRMR = .040.

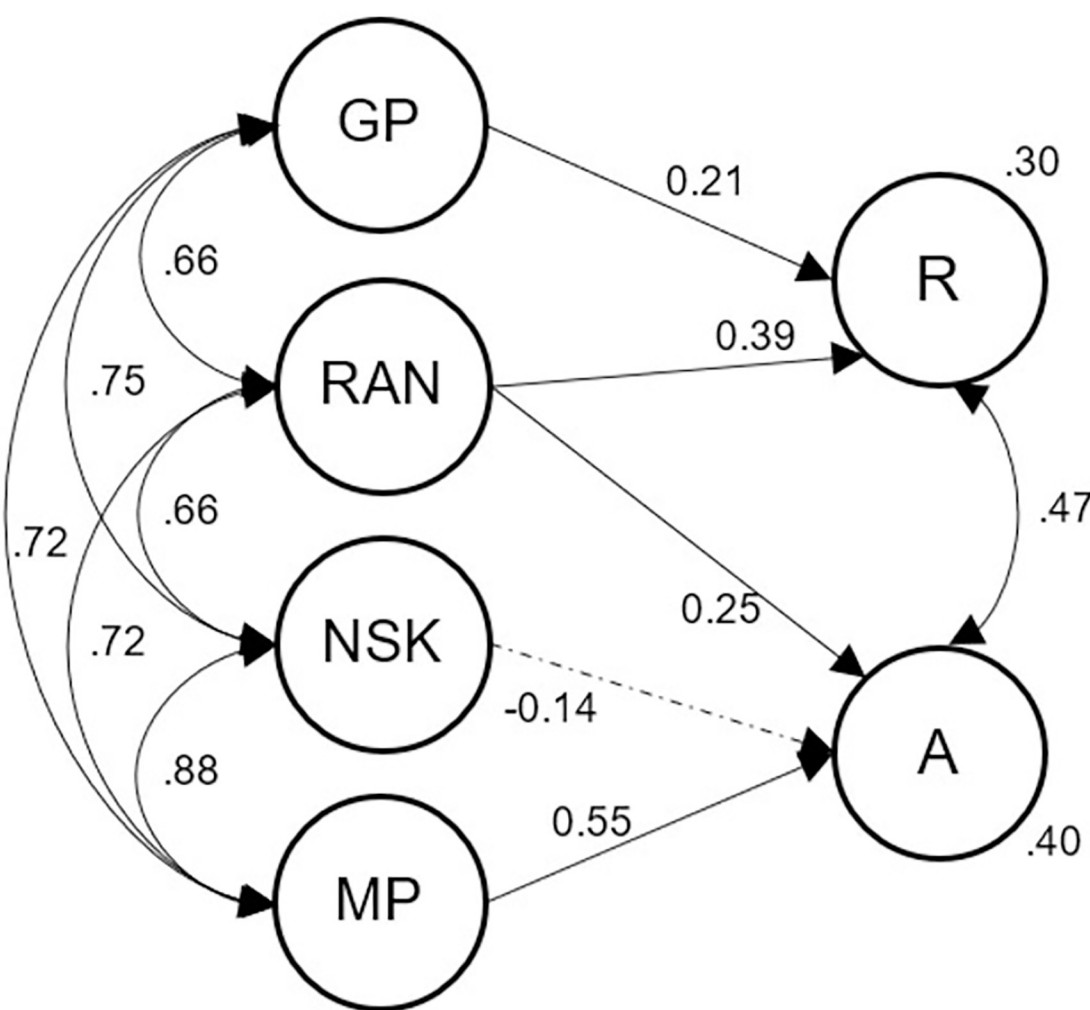

**Fig 3. Model 1: Shared and specific predictors of reading and arithmetic.** GP = Grapheme-phoneme processing; RAN = Rapid automatized naming; NSK = Number system knowledge; MP = Magnitude processing; R = Reading; A = Arithmetic. Dashed lines indicate non-significant paths.

Grapheme-phoneme processing was a significant predictor of reading ($p$ = .001), while RAN was a shared predictor of both reading ($p$ < .001) and arithmetic ($p$ = .004). Magnitude comparison was predictive of arithmetic ($p$ = .003), but number system knowledge was not ($p$ = .343). This was surprising given previous findings linking counting and/or number knowledge to arithmetic [27, 48, 56]. This unexpected finding is possibly explained by the high correlation between number system knowledge and magnitude processing ($r$ = .88), with magnitude processing as the main predictor of arithmetic. Indeed, an additional analysis confirmed that number system knowledge was a significant predictor of arithmetic on its own ($p_{NSK}$ < .001), but did not explain unique variance, when magnitude processing was considered. Overall, 30% of reading variance and 40% of arithmetic variance were explained. Including all predictors for each outcome variable did not change the amount of explained variance (reading: 29%; arithmetic: 42%).

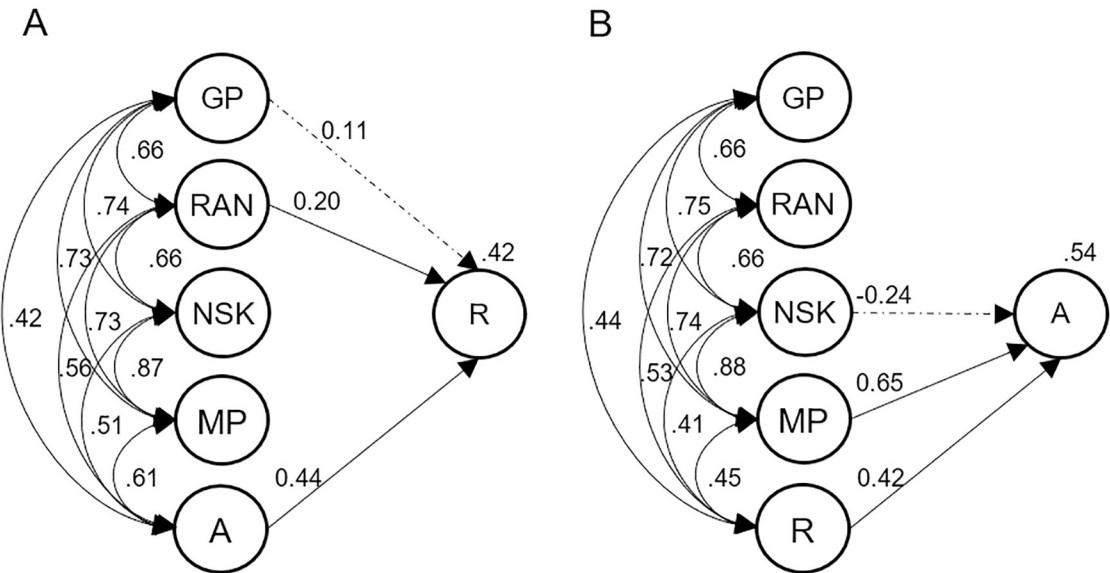

**Fig 4. Model 2 and Model 3: Predicting specific reading (left) and arithmetic (right) variance.** (A) R = Reading. (B) A = Arithmetic. (A & B) GP = Grapheme-phoneme processing; RAN = Rapid automatized naming; NSK = Number system knowledge; MP = Magnitude processing. Dashed lines indicate non-significant paths.

## Model 2 and 3—Cognitive predictors explaining unique variance of reading and arithmetic

Two additional structural equation models were run to investigate which predictors explained specific variance in reading (Model 2) or arithmetic (Model 3). This was achieved by including the second domain of academic learning as an additional predictor, to account for all variance shared between reading and arithmetic. Grapheme-phoneme processing, RAN, and arithmetic were entered as predictors of reading in Model 2. Number system knowledge, magnitude processing, and reading were entered as predictors of arithmetic. The same modifications as in Model 1 were applied and fit indices indicated good fit, reading: $X^2(149) = 399.154$, $p < .001$, $X^2/df = 2.68$; CFI = 0.951; TLI = 0.937; RMSEA = .044 (90% CI = .038–.049); SRMR = .040; arithmetic: $X^2(149) = 400.223$, $p < .001$, $X^2/df = 2.69$; CFI = 0.951; TLI = 0.937; RMSEA = .044 (90% CI = .039–.049); SRMR = .040. In Model 2 (Fig 4A), RAN ($p = .002$) and arithmetic ($p < .001$) were significant predictors of reading ($R^2 = .42$), while grapheme-phoneme processing was marginally significant ($p = .052$). In Model 3 (Fig 4B), 54% of variance in arithmetic was accounted for by magnitude comparison ($p = .001$) and reading ($p < .001$). Like in Model 1, number system knowledge was no unique predictor of arithmetic ($p = .137$). After excluding magnitude processing however, number system knowledge again predicted arithmetic ($p < .001$), but the amount of explained variance decreased ($R^2 = .47$). Including RAN in Model 3, which was a significant predictor in Model 1, did not change the results ($p_{RAN} = .356$; $R^2 = 52\%$). The amount of variance explained in Models 2 and 3 changed only minimally in less parsimonious Models using all predictors ($R^2$: .44 for reading; .54 for arithmetic).

## Model 4—Predictors explaining covariance of reading and arithmetic

In a fourth model we aimed to identify predictors explaining variance in skills shared between reading and arithmetic ("covariance"-variable). Again, we estimated a theory-based

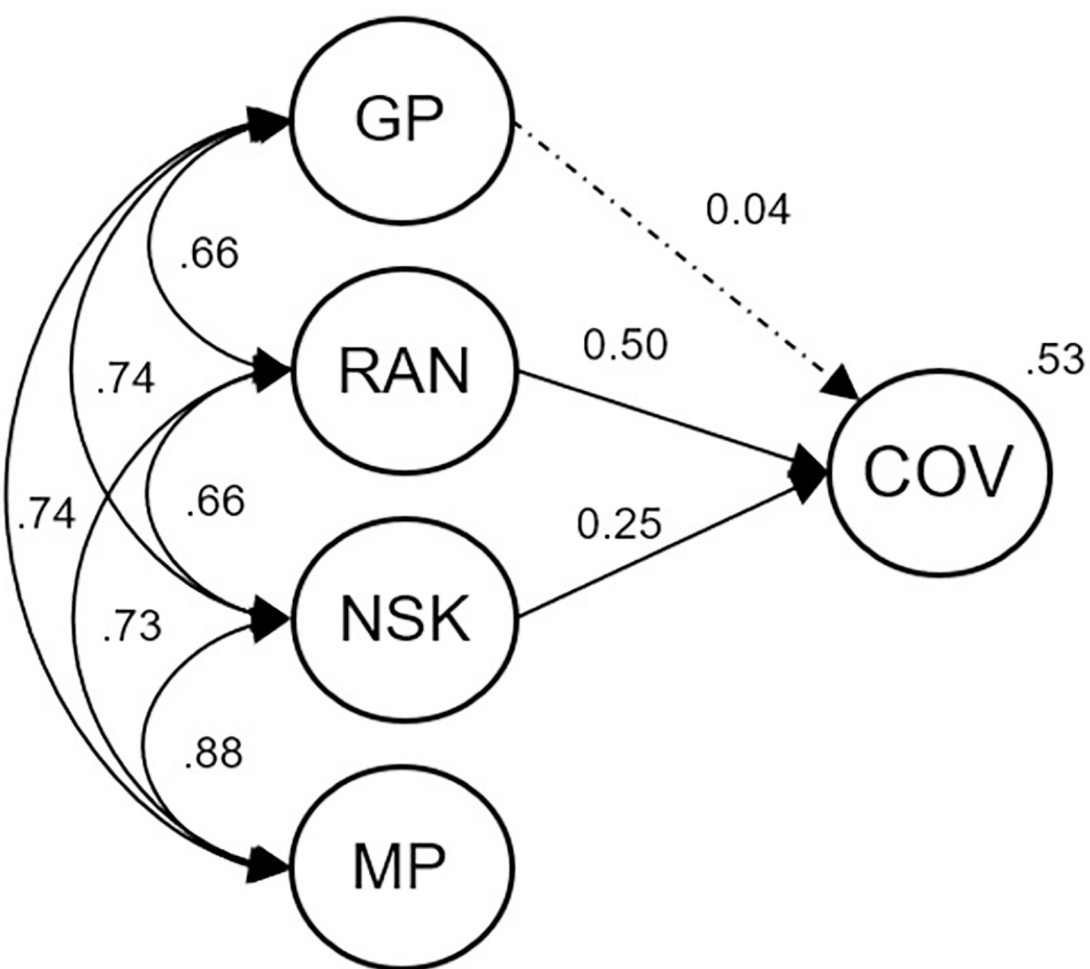

**Fig 5. Model 4: Predicting skills shared between reading and arithmetic.** GP = Grapheme-phoneme processing; RAN = Rapid automatized naming; NSK = Number system knowledge; MP = Magnitude processing; COV = Covariance. Dashed lines indicate non-significant paths.

parsimonious model (Fig 5). We included grapheme-phoneme processing, RAN, and number system knowledge as potential predictors. The links between error variances were maintained from Model 1. We also included links between reading at beginning and end of 2$^{nd}$ grade, as well as arithmetic at beginning and end of 2$^{nd}$ grade, indicating domain-specific components which are not reproduced in the latent covariance-variable. The model seemed to fit the data well, $X^2(151) = 417.29$, $p < .001$, $X^2/df = 2.76$; CFI = 0.948; TLI = 0.934; RMSEA = .045 (90% CI = .040–.050); SRMR = .042. RAN ($p < .001$) and number system knowledge ($p = .006$) were both predictive of reading and arithmetic covariance in Model 4, explaining 53% of variance, i.e., the higher those skills the higher reading **and** arithmetic levels. We reran the model without number system knowledge to investigate whether the lack of unique prediction of grapheme-phoneme processing was due to its shared properties with number system knowledge ("symbol knowledge"). After this exclusion, grapheme-phoneme processing predicted reading and arithmetic covariance ($p = .004$), with no essential change in explained variance ($R^2 = .52$). Including all variables as predictors would again only slightly change $R^2$ ($R^2 = .55$).

## Discussion

The goal of the present study was to investigate cognitive mechanisms driving associations and dissociations of reading and arithmetic at the beginning of formal schooling. Investigating early development of academic skills is particularly important as this is when the foundations of the neuro-cognitive networks for reading and arithmetic (and their overlaps) are established. The predictors we implemented in our study design are generally seen as domain-specific [21, 22, 84], but this view was recently challenged by a number of studies [27, 29, 44–46]. In the current study we tested four structural equation models to see 1) which predictors account for variance in reading and/or arithmetic (Model 1), 2) which predictors explain variance specific to reading (Model 2) or arithmetic (Model 3) and thereby providing potential explanations for dissociations between those skills, and 3) which predictors explain variance shared between reading and arithmetic (Model 4). Taking the results of all four structural equation models into account provides a fine-grained picture of the pattern of predictions. Interestingly, the current analyses are rather supportive of the domain-specificity view, with a number of critical adaptations towards cross-domain associations. In the following, we will discuss the evidence first separately for reading and arithmetic and then for their covariance.

### Predictors of reading

In line with our expectations and earlier literature [24, 25, 27, 29, 47], we found grapheme-phoneme processing and RAN to be longitudinal predictors of reading (Model 1). Interestingly, when arithmetic was included as a predictor in order to control for all variance shared between the two academic skills (Model 2), grapheme-phoneme processing just about missed standard levels of significance as a predictor ($p$ = .052), showing that mechanisms relevant for arithmetic partly account for the association between grapheme-phoneme processing and reading. We will discuss potentially overlapping mechanisms in the section on predictors of covariance below.

RAN turned out to account for unique variance in reading when arithmetic (and thus the variance shared between the two skills) was accounted for (Model 2). We see two potential explanations for this effect: First, reading aloud as assessed in our word and nonword reading tasks shares a naming component with RAN, which was not required in the arithmetic fluency measure (written responses). Second, sequential visual-verbal processing is an inherent component of reading, whereas arithmetic is not always done in a sequential format as in our fluency task. It remains to be seen whether RAN also shows a significant relation with untimed arithmetic or more complex math reasoning. In summary, RAN may not be domain-specific to reading and is certainly associated with mechanisms overlapping between the two academic skills, but it is perhaps still more relevant for reading than for arithmetic. An important implication of this finding is that individuals with RAN-deficits are likely to experience more marked problems in reading than in arithmetic.

Also in line with our expectations, neither number system knowledge nor magnitude processing showed specific relations with reading (Model 2). Efficient processing of numbers and magnitudes does not seem to be particularly relevant for learning to read words, nonwords and sentences fluently.

### Predictors of arithmetic

Again, our assumptions that number system knowledge, magnitude comparison, and RAN should predict arithmetic performance while grapheme-phoneme processing should not be of particular relevance were largely met. The deviation from our expectations was that number system knowledge (extracted from verbal-visual number identification and a purely verbal

number word successor task) was no significant predictor of arithmetic performance (Model 1 & 3), not even when differences in reading skills were not considered (Model 1). This negative finding was surprising given that earlier research found knowing and understanding the complexities of the number (word) system to be a main predictor of later arithmetic, even when magnitude comparison was considered [45, 70, 85]. While we observed positive bivariate correlations between the compounds of number system knowledge and arithmetic, the effect was insignificant in the structural equation models 1 and 3, with an even negative beta weight which might be indicative of a (minor) suppressor effect. Given the high correlations between the latent variables number system knowledge and magnitude processing (.87–.88), it seems that these two components assessed similar and clearly overlapping constructs of basic numerical processing. This is further confirmed by the finding that excluding magnitude processing resulted in a significant positive effect of number system knowledge on later arithmetic. In the full model, number system knowledge therefore showed some suppressor effect, meaning that a positive association between number system knowledge and arithmetic was accounted for by magnitude processing, while number system knowledge accounted for criterion-irrelevant variance of magnitude comparison.

Grapheme-phoneme processing was no relevant predictor of arithmetic not even when variance in reading was not controlled for (Model 1). As pointed out earlier, the verbal component relevant for arithmetic does not necessarily involve access to sublexical phonology as it is assessed in our grapheme-phoneme processing variable. Awareness of the phonemic structure of words is not necessary for arithmetic, while phonological processing on the word level plays an important role for number fact storage. We assume that earlier findings on the association of phonological awareness and arithmetic [41] mostly reflect this more general verbal component of number fact storage. This would imply that children who have problems with explicit phoneme awareness but not general phonology would be at risk for deficits in reading, while their arithmetic performance might be fully intact. But of course, many children experience broader deficits in phonology, including explicit phonological awareness [86–88]. In such cases, both reading and arithmetic development (e.g., number fact knowledge) might be impaired, constituting the frequently observed comorbidity of learning disorders.

Finally, RAN was traditionally seen as a domain-specific predictor of reading, but in line with earlier research [46, 52, 53], we found clear evidence that it is associated to mechanisms that are relevant for reading as well as arithmetic. It predicted both skills in Model 1 and was the numerically strongest predictor of the covariance variable in Model 4. Thus, RAN will be discussed in detail in the next section.

## Predictors of the covariance between reading and arithmetic

As expected and in line with earlier evidence [28, 29, 45, 47], RAN was a significant predictor of the covariance in reading and arithmetic (Model 4). The mechanisms underlying RAN are as yet a subject of debate [51, 89–91]. The different perspectives largely converge in assuming that speeded sequential naming involves the timely integration of visual and verbal skills and simultaneous processing of multiple stimuli presented serially [49, 51]. Arguably, these skills are relevant for reading as well as arithmetic, especially in the task formats used in the current study. It is highly likely that overlapping speed requirements for most of our reading and arithmetic tasks contributed to the association of their covariance with RAN. Note, however, that RAN predicted arithmetic even when magnitude processing was included as a predictor (Model 1), which was also assessed in terms of speed measures.

A second cognitive component that is involved in fluent reading as well as arithmetic is processing of arbitrary visual-verbal symbols in terms of letters and Arabic digits. In the current

study, symbolic processing was part of the latent variables grapheme-phoneme processing and number system knowledge, which were tightly correlated with each other ($r$ = .74–.75). It was actually number system knowledge that predicted the covariance variable, while grapheme-phoneme processing only became significant when number system knowledge was excluded from the model. This evidence aligns well with earlier studies also finding either letter or number knowledge, but not both as predictors of covariance between reading and arithmetic [29, 44], probably because the two competencies are too closely related with each other to show independent contributions.

Our assumption that a model without magnitude processing as a predictor of covariance would fit the data well was also confirmed, even though this latent variable also involved a symbolic condition requiring efficient processing of Arabic digits. We conject that this component is more strongly reliant on the magnitudes represented by simple single digit numbers and is thus specifically related to arithmetic, but not reading.

## Associations and dissociations of reading and arithmetic

In summary, our findings support the view that reading and arithmetic are associated with distinct, domain-specific predictors: Grapheme phoneme processing is related to reading, but not arithmetic, while magnitude processing is related to arithmetic, but not reading (Models 1–3). RAN also shows a specific association with reading above and beyond its contribution to the covariance of the two academic skills (Model 2). Serial visual-verbal processing as required by RAN and efficient processing of visual-verbal symbols are components that are involved in both reading and arithmetic.

The similarities in processing mechanisms between reading and arithmetic are elegantly represented in the obvious analogies between the Lexical Quality Hypothesis of reading [5] and the Triple Code Model of numerical cognition ([6]; Fig 1). Efficient integration of visual, verbal and semantic information is crucial for both reading and arithmetic. The general ability to form and use associations between these sources of information efficiently (as, for instance, reflected by RAN or working memory, which was not assessed in the current study) would impact on reading as well as arithmetic. We speculate that the high relevance of building such associations for both academic skills may contribute to the lack of clear neuronal differences in studies comparing dyslexia and dyscalculia ([11–13], but see [14]).

Importantly, however, these complex integrational processes may fail for different reasons: A specific component of reading alphabetic orthographies is that associations need to be built on the level of sublexical phonology. As pointed out earlier, children who have problems with this particular phonological competence may experience difficulties to build up and retrieve lexical representations for written words, while their numerical and arithmetic processing may be unaffected. On the other hand, numerical processing requires a very specific type of semantic information, namely understanding numerosities. Individuals who experience deficiencies in this evolutionary based skill [92–94] would be expected to show problems in establishing fully integrated and automatically retrievable numerical representations, but might well be able to build up high quality lexical representations for written words. This line of argument corresponds well with current causal models of dyslexia in terms of a phonological deficit [95, 96] and dyscalculia in terms of a deficit in basic numerical processing [97–99] and with empirical evidence finding clear dissociations between the cognitive profiles of the two learning disorders [100–102]. Having said that, a phonological deficit that extends beyond phonological awareness may well impact on arithmetic development (e.g., number fact knowledge) and deficits in specific subcomponents of numerical processing (e.g., processing of Arabic digits). Overall, this analysis shows that the similarities and differences between the Lexical Quality

Hypothesis of reading [5] and the Triple Code Model of numerical cognition [6] provide a useful theoretical background to better understand associations and dissociations between reading and arithmetic and the heterogeneity of learning disorders.

## Limitations and future studies

Statistical models strongly depend on the variables assessed in a particular study. The heterogeneity between study designs in the field of predicting reading and/or arithmetic is likely to explain many of the differences in findings. In the current study, we were particularly interested in the association of standard domain-specific predictors of reading and arithmetic. Domain-general predictors like nonverbal reasoning, working memory or other executive functions were not considered. This will be important in future studies in order to get a complete picture of the mechanisms underlying associations and dissociations of reading and arithmetic.

The current longitudinal study focused on a relatively young age group (before/at school entry to Grade 2) in order to assess a period in which the foundational skills of reading and arithmetic are typically established. A latent variable modeling approach seemed particularly important for this age group as it can reduce the impact of measurement error. However, we had to combine assessments from two time points for predictors as well as criterion measures. Future studies should aim to investigate developmental changes in the predictive patterns in more detail. Furthermore, even though the association between reading and arithmetic appears to be quite stable throughout development [103–106], it will be important to investigate predictive patterns in older age groups.

As the Austrian kindergarten curriculum does not provide explicit instruction for letters and digits, about 15% of our sample were not yet familiar with all digits that were presented in the RAN-digits task at the first time point. Alphanumeric RAN is generally considered a better predictor than non-alphanumeric RAN [107, 108] so we wanted to include it in our prediction measures. We were actually positively surprised that the vast majority of children were well able to perform the task at t1 and no problems were any longer evident at t2 (beginning of Grade 1), even though children had just about entered formal schooling. Note that the RAN-digits score was imputed for those children who could not name all digits, ensuring that their RAN score was not compromised by low digit knowledge.

Children´s young age at the beginning of our longitudinal study may explain why we were unable to assess fully distinct constructs, as reflected in high correlations between the latent predictor variables. Other studies used similar latent predictors in a comparably young sample [44, 47, 56]. Amland et al. [44] measured phonological awareness, number, and letter knowledge, but did not report correlations between latent variables. Koponen et al. [56] assessed phonological awareness, RAN, and counting, but only reported correlations between RAN and counting, which were high as well (.62). Another study [47] assessed phonological awareness, RAN, and counting, but only included RAN and counting as latent variables, which showed a moderate correlation of .27. Further studies should aim to investigate to what extent tasks can be improved to assess the constructs more clearly in young age groups.

## Conclusion

The results of the current longitudinal study following the early development of reading and arithmetic and their covariance corroborate and expand the recent cognitive literature on potential factors underlying the remarkable association between those academic skills. Grapheme-phoneme processing and RAN were confirmed as predictors of later reading, while magnitude comparison as well as RAN were predictive of arithmetic. Skills shared between reading

and arithmetic were predicted by RAN and number system knowledge. We explained our findings in terms of similarities and differences between the Lexical Quality Hypothesis [5] and the Triple Code Model of numerical cognition [6], which both rely on a tight integration of visual, verbal, and semantic information, but nevertheless entail specific subcomponents (explicit phonological awareness for reading and understanding numerosities for arithmetic) that can explain dissociations between the two academic skills. We propose that this perspective can also provide a useful theoretical background to investigate the comorbidity between learning disorders.

## Acknowledgments

We want to thank Anna Exel, Victoria Wagner, and Anna Kaltenberger for their help in data collection. We thank all schools and especially the children and their parents for their participation in this study.

## Author Contributions

**Conceptualization:** Viktoria Jöbstl, Karin Landerl.

**Data curation:** Viktoria Jöbstl.

**Formal analysis:** Viktoria Jöbstl.

**Funding acquisition:** Pia Deimann, Ursula Kastner-Koller, Karin Landerl.

**Investigation:** Viktoria Jöbstl, Anna F. Steiner.

**Methodology:** Viktoria Jöbstl, Karin Landerl.

**Project administration:** Viktoria Jöbstl, Anna F. Steiner, Pia Deimann, Ursula Kastner-Koller, Karin Landerl.

**Resources:** Pia Deimann, Ursula Kastner-Koller, Karin Landerl.

**Supervision:** Pia Deimann, Ursula Kastner-Koller, Karin Landerl.

**Validation:** Viktoria Jöbstl, Karin Landerl.

**Visualization:** Viktoria Jöbstl.

**Writing – original draft:** Viktoria Jöbstl, Karin Landerl.

**Writing – review & editing:** Viktoria Jöbstl, Karin Landerl.

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
