## [Decision Letter · Decision Letter 0]

31 Jan 2023

PONE-D-22-31656AB3 - associations and dissociations of reading and arithmetic: Is domain-specific prediction outdated?PLOS ONE

Dear Dr. Jöbstl,

Thank you for submitting your manuscript to PLOS ONE. After careful consideration, we feel that it has merit but does not fully meet PLOS ONE’s publication criteria as it currently stands. Therefore, we invite you to submit a revised version of the manuscript that addresses the points raised during the review process.

We look forward to receiving your revised manuscript.

Kind regards,

Madelon van den Boer

Academic Editor

PLOS ONE

Journal Requirements:

Additional Editor Comments:

Your manuscript has been reviewed by two experts in the field. I believe their comments are quite clear and can be addressed in a revision. From my own reading of the manuscript I would like to add a few additional/related issues.

1. The abstract is quite long. Although it might not exceed the word limit, I believe a somewhat shorter abstract would be preferable to attract readers.

2. The specific developmental period under study (from preschool to Grade 2) could receive some more attention in the Introduction. Why is it important to start before official schooling? What is so particular about early development in reading/mathematics?

3. If RAN digits was discontinued in part of the sample, can it still be used as a predictor? Instead of naming speed, RAN digits might become an ability scale?

4. Why did the authors combine several reading/math scores into one score? Why is this approach different for the predictors? It seems to me that different aspects of reading/math might relate to different predictors. I would prefer a latent variable approach for the outcome variables.

Reviewers' comments:

Reviewer's Responses to Questions

**Comments to the Author**

1. Is the manuscript technically sound, and do the data support the conclusions?

Reviewer #1: Yes

Reviewer #2: Yes

2. Has the statistical analysis been performed appropriately and rigorously? 

Reviewer #1: Yes

Reviewer #2: Yes

3. Have the authors made all data underlying the findings in their manuscript fully available?

Reviewer #1: Yes

Reviewer #2: No

4. Is the manuscript presented in an intelligible fashion and written in standard English?

Reviewer #1: Yes

Reviewer #2: Yes

5. Review Comments to the Author

Reviewer #1: Review of paper: AB3 - associations and dissociations of reading and arithmetic: Is domain-specific prediction outdated?

The current study aimed at re-evaluating the domain-specificity of established predictors of reading and arithmetic, and investigated the prediction of variance shared between the two academic skills and unique to one skill. The study was performed among 885 German-speaking children before and/or at the onset of formal schooling. The result exhibited a differentiated perspective of cognitive predictors: Reading and arithmetic require tight networks of visual, verbal, and semantic information, as reflected by RAN. In addition, domain-specific cognitive components were also found.

This study is important and interesting and provides useful information to explain associations and dissociations between reading and arithmetic performance.

The paper is well written but before publishing there are several issues to address:

1. The introduction is well written but I think information regarding the common cognitive mechanizes such as working memory and Executive functions is missing, it is mentioned in the discussion but it is important to give some information in the introduction due to the large amount of research which was performed on these topics and are relevant for this paper.

2. In the method: some of the test have low alpha – is there a good expiation for these results?

3. It is important to explain why the data from t1+t2 and t3+t4 were merged together and not examined separately.

4. There is no information regarding how the reading and arithmetic variables were created for t3 and t4 which measures were entered? Accuracy? Fluency? Decoding? Comprehension? And the same in the arithmetic domain? It is important to understand why they were are merged together and not examined separately.

5. The main finding is regarded to the RAN, there are many theories which try to explain what RAN represents, which are not presented in this paper one is that is represents speed of processing or speed of retrieval in general. While most of the reading and math tests were timed it is not surprising theta they were highly connected to the RAN test. This is an important point which need to be explained widely in the discussion.

Reviewer #2: Thank you for the opportunity to review this manuscript.

The present study investigated the prediction of variance shared between the two academic skills and unique to one skill in a sample of 885 German-speaking children.

Overall, manuscript has clear structure and is mostly well written. I believe that this paper has potential to make a nice contribution to research However, there are several issues that should be addressed

Introduction:

On lines 200-204: ”While many children start naming simple Arabic digits by the age of two or three

years [53, 54], this knowledge becomes useful for arithmetic only when children associate

the symbol not only with the corresponding number word, but also with its numerical meaning .”

The early number concept development starts from learning the small number words and their quantitative meaning and later the quantity-digit and number word – digit association. Three years old don’t typically master the Arabic digits yet (E.g., Benoit et al., 2013). Please rewrite this part.

Method:

Using term “screening” in method section is a bit misleading, because none of the children were selected to leave out or were they? If not, I would use word “assessment” instead of screening (for example on lines 304 and 306).

Please provide more information when digital versions were used and when paper-pencil tasks. Add also the information whether the correspondence between different test versions have been examined earlier.

Please clarify, what is meant by: “Some cognitive tasks had parallel versions, which differed in item order (phonological awareness, symbolic magnitude comparison) and for some tasks also in item

sets (letter identification, number identification, successor knowledge, t1 non-symbolic

magnitude comparison). With one exception (t1 letter identification – see task section for

details) mean item difficulties were similar and scores were comparable between versions.”

It’s not clear whether same items/test versions were used for all participants at certain time point and parallel versions across the time or whether parallel versions were used within the same assessment point. Thus, please clarify how parallel versions were used, as well as how the comparability between versions was verified.

RAN digits were assessed already in kindergarten, although it’s described that “Kindergarten activities vary greatly, and mostly focus on social and language skills, while activities including letters or numbers are highly exceptional”. Please provide rationale for that and discuss the limitations of this solution.

Results

Latent variables for reading and arithmetic were constructed from variables measuring skills in the beginning and end of 2nd grade. Please, provide rationale for that. I think that during the 2nd grade clear development in both skills typically take place, and also the changes in the order of the children are possible (although here the correlation in reading was very high). Now the shared skill levels across the assessment points were selected for the outcome. Please provide rationale for that and discuss the possible impacts of selected approach.

6. PLOS authors have the option to publish the peer review history of their article (what does this mean?). If published, this will include your full peer review and any attached files.

Reviewer #1: **Yes: **Shelley Shaul

Reviewer #2: No

---

## [Author Response · Author response to Decision Letter 0]

7 Mar 2023

PONE-D-22-31656

AB3 - associations and dissociations of reading and arithmetic: Is domain-specific prediction outdated?

We are grateful for the overall positive evaluation of our manuscript and the helpful feedback provided by the editor and reviewers. In the following paragraphs, we will address each comment (highlighted in italic and grey) in detail. The comments were most helpful in creating a more transparent manuscript. We have responded to each comment separately, retaining the original order. Presented line numbers (e.g., lines 52-55) refer to lines of the revised manuscript (without markings).

Journal Requirements:

**RESPONSE: We controlled adherence of all requirements and made changes accordingly. 

**RESPONSE: We included the necessary information and a link to the Open Science Framework repository to grand access to the data set, which will be open to public upon acceptance (https://osf.io/h89ue/?view_only=64373626a129449fa3baa08c791b8305)

**RESPONSE: Data will be made public upon acceptance – see above. 

Additional Editor Comments:

Your manuscript has been reviewed by two experts in the field. I believe their comments are quite clear and can be addressed in a revision. From my own reading of the manuscript I would like to add a few additional/related issues.

1. The abstract is quite long. Although it might not exceed the word limit, I believe a somewhat shorter abstract would be preferable to attract readers.

**RESPONSE: We have now shortened the abstract from 296 to 233 words. While doing so, we did not change the method and result sections too much, to keep the conclusions transparent. 

2. The specific developmental period under study (from preschool to Grade 2) could receive some more attention in the Introduction. 

2.a. Why is it important to start before official schooling? 

**RESPONSE: This decision was made for both theoretical and methodological reasons: Assessing predictors prior to the start of formal schooling reduces the impact of confounding effects due to interdependences (phonological awareness ⇄ reading). Early identification of potential strengths and weaknesses also allows for early intervention and individualized instruction. This is now explained in the introduction in the "current study" section (lines 256-260).

2. b. What is so particular about early development in reading/mathematics?

**RESPONSE: One of our study’s main objectives was to identify the cognitive processes shared between networks of reading and arithmetic. We believe that the early period of academic learning is crucial as this is the time foundations of are established. We acknowledged the significance of this period in several instances (e.g., lines 110-113; 252-254; 652-654; …). For example, we noted that “In order to reveal the mechanisms underlying the (co-)development of academic skills, it is particularly important to focus on the early school years when foundational skills of reading and arithmetic as well as their co-occurrence are established.”; lines 110-113). 

As we do not investigate changes in reading and arithmetic over time, we further state in the discussion: “Furthermore, even though the association between reading and arithmetic appears to be quite stable throughout development [104–107], it will be important to investigate predictive patterns in older age groups” (lines 813-815).

3. If RAN digits was discontinued in part of the sample, can it still be used as a predictor? Instead of naming speed, RAN digits might become an ability scale?

**RESPONSE: We have discussed this issue extensively. We wanted to assess RAN-digits) as it is often a better predictor than non-alphanumeric RAN (Araújo et al. 2017, McWeeny et al., 2022). Still, we were well aware that some children might not yet be sufficiently familiar with some digits in order to name them fluently as required for RAN. While other preschool studies typically do not report on RAN accuracy (e.g., Cirino et al., 2018), we decided to not even run the task if it was clear that at least one digit was not familiar to the child. Thus, for those children that have a RAN-digits score, we are certain, that the task measures naming speed rather than simple digit knowledge. For the 98 children who were unfamiliar with at least one digit, we decided to impute a naming speed score. As a matter of fact, as part of our preliminary analyses, we had imputed RAN-digits (t1) using RAN-digits (t2) and RAN-objects (t1 & t2) as references to ensure that the imputed score reflects naming speed. This approach did not differ in prediction pattern from the model we report in the paper. We now expanded on this possible limitation in the discussion (lines 816-824). 

4. Why did the authors combine several reading/math scores into one score? Why is this approach different for the predictors? It seems to me that different aspects of reading/math might relate to different predictors. I would prefer a latent variable approach for the outcome variables.

**RESPONSE: Reading was assessed by three fluency-based tasks: reading aloud of (1) words and (2) nonwords as quickly as possible and (3) silent sentence reading fluency (with minor comprehension requirements as children had to decide whether or not the sentence was semantically adequate). So, the main construct that was assessed was reading fluency. Arithmetic was also assessed by three tasks at both time points: (1) addition and (2) subtraction fluency and (3) more complex written calculations without time limit. So again, we focused on one construct, namely mental calculations. 

We had initially planned to use a latent variable approach based on the different tasks, however, statistical limitations prevented us from doing so. We have now included details on why we used the reported approach in the method section (lines 510-528): 

“The criterion measures reading and arithmetic were assessed with three different tasks per time point (t3/t4). A model using the three tasks as indicators for latent variables resulted in non-positive definite covariance matrices. At first, we assumed that this was because the assumption of local independence was not met, i.e., because of closer format overlap of two out of the three tasks (word and nonword reading had the same format of reading aloud for one minute whereas sentence reading was silent and children had to mark semantic correctness; addition and subtraction fluency had the same task format, whereas numerical operations consisted of more complex arithmetic problems with less focus on speed). We attempted to address this issue by averaging z-transformed scores of the two similar tasks (word and nonword reading; addition and subtraction fluency) and using this mean and the score of the third task as indicators of a latent variable. However, this again resulted in a non-positive definite covariance matrix, indicating that the observed variables were not related to each other in a way that could be explained by the hypothesized latent variable structure. As both reading and arithmetic tasks showed reasonable to high stability across the two assessments (reading: 849-.888; arithmetic: .550-.674), we chose to reduce the complexity of the model and use a similar approach as for the predictors by combining the two time points (t3/t4): We z-transformed all tasks separately and averaged the three reading and the three arithmetic tasks. The mean reading and arithmetic scores of the two time points were then used as indicators of latent variables for reading and arithmetic.”

Reviewers' comments:

Reviewer's Responses to Questions

Comments to the Author

1. Is the manuscript technically sound, and do the data support the conclusions?

Reviewer #1: Yes

Reviewer #2: Yes

2. Has the statistical analysis been performed appropriately and rigorously? 

Reviewer #1: Yes

Reviewer #2: Yes

3. Have the authors made all data underlying the findings in their manuscript fully available?

Reviewer #1: Yes

Reviewer #2: No

4. Is the manuscript presented in an intelligible fashion and written in standard English?

Reviewer #1: Yes

Reviewer #2: Yes

5. Review Comments to the Author

Reviewer #1: Review of paper: AB3 - associations and dissociations of reading and arithmetic: Is domain-specific prediction outdated?

The current study aimed at re-evaluating the domain-specificity of established predictors of reading and arithmetic, and investigated the prediction of variance shared between the two academic skills and unique to one skill. The study was performed among 885 German-speaking children before and/or at the onset of formal schooling. The result exhibited a differentiated perspective of cognitive predictors: Reading and arithmetic require tight networks of visual, verbal, and semantic information, as reflected by RAN. In addition, domain-specific cognitive components were also found.

This study is important and interesting and provides useful information to explain associations and dissociations between reading and arithmetic performance.

The paper is well written but before publishing there are several issues to address:

1. The introduction is well written but I think information regarding the common cognitive mechanizes such as working memory and Executive functions is missing, it is mentioned in the discussion but it is important to give some information in the introduction due to the large amount of research which was performed on these topics and are relevant for this paper.

**RESPONSE: We appreciate your feedback. Although domain-general predictors were not included in our analyses, they play a significant role in the reading and arithmetic literature. We have now added information on domain-general predictors to the introduction of our manuscript (lines 115, 116) to provide a comprehensive overview. 

2. In the method: some of the test have low alpha – is there a good expiation for these results?

**RESPONSE: Cronbach´s alpha was higher than .70 for most tasks with the exception of non-symbolic magnitude comparison at t2 where it was only minimally lower (.68) and number identification at t2 (.51 and .55)). As we are aware of these limitations, we only used latent variables to partly account for error variance, which we now referred to in the analysis plan (line 499-504). See also our response to the next comment.

3. It is important to explain why the data from t1+t2 and t3+t4 were merged together and not examined separately.

**RESPONSE: 

T1 and T2: As children were very young, we wanted to create latent variables to account for measurement error and measure factors as accurately as possible. We only used two tasks per time point and construct, which were only moderately correlated and partly showed low to moderate reliability. Therefore, we used data from two time points, resulting in four indicators per factor, similar to studies such as Koponen et al. 2013 or Koponen et al. 2016.

T3 + T4: See our response to question 4 of the editor. 

We agree that this should be explained in more detail and now provide more information in the analysis plan (lines 510-528).

4. There is no information regarding how the reading and arithmetic variables were created for t3 and t4 which measures were entered? Accuracy? Fluency? Decoding? Comprehension? And the same in the arithmetic domain? It is important to understand why they were are merged together and not examined separately.

**RESPONSE: As explained in the Methods section (lines 474 and 479 for reading and 490 for arithmetic), we scored the number of correct items for word and nonword reading and also for addition and subtraction fluency. For sentence reading we subtracted the number of incorrect responses from the number of correct responses, in order to account for potential guessing. For written calculations the number of correct items were scored. The reviewer is correct that our explanations as to how these tasks were combined into one latent variable for reading and one for arithmetic was insufficient and was now expanded (lines 510-528). 

5. The main finding is regarded to the RAN, there are many theories which try to explain what RAN represents, which are not presented in this paper one is that is represents speed of processing or speed of retrieval in general. While most of the reading and math tests were timed it is not surprising theta they were highly connected to the RAN test. This is an important point which need to be explained widely in the discussion.

**RESPONSE: The relationship between RAN and reading has been studied extensively, but the exact driving components are not yet clear. Various theories have been proposed, including processing speed, phonological, and orthographic processing, but even after controlling for these factors, RAN remained a significant predictor of reading (for a review see Kirby et al., 2010). Overall, serial and verbal components of RAN-tasks seem to play a role (Georgiou et al. 2013, Protopapas et al., 2013) while the exact nature of what RAN measures is still unclear. 

We agree with the reviewer that the prediction of RAN for covariance is perhaps partly due to overlapping speed requirements. But interestingly, RAN accounted for variance in reading even when (speeded) arithmetic skills were entered into model 2. This association cannot be explained by processing speed. Furthermore, magnitude processing was also a speed measure, but showed only a specific association with arithmetic, but not with reading. We now included an additional section in the discussion, where we expanded on possible factors driving the RAN-reading and the RAN-arithmetic relationship (lines 188-192; 734-743).

Reviewer #2: Thank you for the opportunity to review this manuscript.

The present study investigated the prediction of variance shared between the two academic skills and unique to one skill in a sample of 885 German-speaking children.

Overall, manuscript has clear structure and is mostly well written. I believe that this paper has potential to make a nice contribution to research However, there are several issues that should be addressed

Introduction:

On lines 200-204: ”While many children start naming simple Arabic digits by the age of two or three

years [53, 54], this knowledge becomes useful for arithmetic only when children associate

the symbol not only with the corresponding number word, but also with its numerical meaning .”

The early number concept development starts from learning the small number words and their quantitative meaning and later the quantity-digit and number word – digit association. Three years old don’t typically master the Arabic digits yet (E.g., Benoit et al., 2013). Please rewrite this part.

**RESPONSE: Our initial phrasing was misleading/incorrect, as children start using number words but do not reliably associate number words with digits. Thank you for the reference. Benoit et al. (2013) found that three-year-old children are able to associate number words with quantities, but their mapping of number words with digits was at chance level. Consistent mapping of number words to digits was only observed in four-year-olds. We have now rewritten this section:

"The development of the number concept usually starts with learning small number words and quantitative meaning at the age of two or three years (Mix, 2002; Wynn, 1990). By the age of four, children begin to master the associations between quantities, digits, and number words (e.g., Benoit et al., 2013).”

Method:

Using term “screening” in method section is a bit misleading, because none of the children were selected to leave out or were they? If not, I would use word “assessment” instead of screening (for example on lines 304 and 306).

**RESPONSE: We thank the reviewer for the careful reading of our manuscript. As explained in lines 352 and 353, this longitudinal study was part of a larger project aimed to develop and validate a screening tool to identify (and support) children at risk for learning problems early on. Still, apart from this formulation, we are following the reviewer´s suggestion and have replaced “screening” with “assessment”.

Please provide more information when digital versions were used and when paper-pencil tasks. Add also the information whether the correspondence between different test versions have been examined earlier. 

Please clarify, what is meant by: “Some cognitive tasks had parallel versions, which differed in item order (phonological awareness, symbolic magnitude comparison) and for some tasks also in item

sets (letter identification, number identification, successor knowledge, t1 non-symbolic

magnitude comparison). With one exception (t1 letter identification – see task section for

details) mean item difficulties were similar and scores were comparable between versions.” 

It’s not clear whether same items/test versions were used for all participants at certain time point and parallel versions across the time or whether parallel versions were used within the same assessment point. Thus, please clarify how parallel versions were used, as well as how the comparability between versions was verified.

**RESPONSE: We are happy to provide more detailed information on the screening tool and this information is also added to the procedure session (lines 363-375): It was generally the schools that decided whether they wanted to use the digital or the paper-pencil version. During t1, only letter identification, number identification, and non-symbolic magnitude comparison were available as digital versions (presented to 45 % of the sample), while the other tasks were given as paper-pencil versions to all children. During t2, all tasks were available in digital format and paper-pencil format and 70 % of the sample received the digital version. 

Parallel versions were used during the same assessment point. We aimed to use each version with about half of the sample (version A:t1: 48 %; t2: 53 %, version Bt1: 52 %; t2: 47 %). For phonological awareness and symbolic magnitude comparison we created pseudo-parallel forms by creating two pseudorandomized item orders. For letter identification, number identification, successor knowledge, and non-symbolic magnitude comparison (only t1), different item sets were presented in versions A and B. 

As part of preliminary item analysis, we inspected reliabilities, item difficulties and discriminatory power separately for formats (digital, paper-pencil) and parallel versions, and did not observe any systematic differences, apart from letter identification at t1, which is made transparent in the task description section (lines: 399-402): “One t1 item showed a clear ceiling effect. To keep results comparable between parallel versions, the outlier item was excluded and the percent of correct responses was computed for the two parallel versions at t1. The score at t2 was the number of correct responses across all items (max. = 9)”. We further calculated multivariate analyses of variance to compare versions and formats of tasks differing in item sets (t1: letter and number identification, successor knowledge, and non-symbolic magnitude comparison; t2: letter and number identification, and successor knowledge) or presentation format (t1: letter and number identification, and non-symbolic magnitude comparison; t2: all tasks). There was no significant difference for most tasks, with nonsignificant main effects of version (p = .327) and format (p = .430) at t1. At t2, there was a significant effect of version (p < .001), with children receiving higher scores in version A than B (p: letter identification < .001, number identification = .052, successor knowledge < .001). An analysis using all predictors as dependent variables showed that children using version A generally outperformed children using version B, even in tasks not differing in item sets (phonological awareness, RAN-digits, non-symbolic and symbolic magnitude comparison, ps < .024). While there was a main effect of format at t2 (p < .001), this effect was only significant for RAN and successor knowledge (digital > paper-pencil), but not the remaining five tasks (p > .372). RAN and successor knowledge were both instructed by teachers and only data entry was digitized. Therefore, we do not think that item sets (version A or B) or formats impacted results. As all data will be made available, detailed information will be accessible for interested readers as well as a summary of the multivariate analyses which we added to the OSF repository (https://osf.io/h89ue/?view_only=64373626a129449fa3baa08c791b8305). 

RAN digits were assessed already in kindergarten, although it’s described that “Kindergarten activities vary greatly, and mostly focus on social and language skills, while activities including letters or numbers are highly exceptional”. Please provide rationale for that and discuss the limitations of this solution.

**RESPONSE: Please also see our replies to the editor and Reviewer 1. Even though letters and numbers are not systematically introduced in kindergarten, children come across them in their everyday lives. During t2 (beginning of Grade 1), all participants could do the task (even though formal teaching had just about started) and even at t1, about 85 % of the participants were well able to perform RAN digits. For those 98 children who were not yet familiar with all digits that were used in the RAN task, scores were imputed. As explained in our response to the editor, we are confident that this procedure ensures that the t1 digit-RAN score provides a measure of children´s naming speed. As the literature generally shows that alphanumeric RAN is a better predictor than non-alphanumeric RAN (e.g., Araújo et al., 2015), we wanted to include this important predictor in our study.

We now added further explanations on this issue in the discussion (lines 816-824).

Results

Latent variables for reading and arithmetic were constructed from variables measuring skills in the beginning and end of 2nd grade. Please, provide rationale for that. I think that during the 2nd grade clear development in both skills typically take place, and also the changes in the order of the children are possible (although here the correlation in reading was very high). Now the shared skill levels across the assessment points were selected for the outcome. Please provide rationale for that and discuss the possible impacts of selected approach.

**RESPONSE: The explanation and discussion of the rational behind our approach and its impact has now been included in the manuscript (lines 510-528). We hope that these clarifications meet the expectations of the editor (4th comment) and reviewers (reviewer 1: 3rd comment).

**RESPONSE: We uploaded each figure and made changes accordingly.

---

## [Decision Letter · Decision Letter 1]

24 Apr 2023

A-B-3 – associations and dissociations of reading and arithmetic: Is domain-specific prediction outdated?

PONE-D-22-31656R1

Dear Dr. Jöbstl,

Thank you for your clear and thorough revision of the manuscript. The newest version has been reviewed by one of the previous reviewers and myself. We both agree that the issues we have raised have been adequately considered and addressed. We are therefore pleased to inform you that your manuscript has been judged scientifically suitable for publication and will be formally accepted for publication once it meets all outstanding technical requirements.

Kind regards,

Madelon van den Boer

Academic Editor

PLOS ONE

Additional Editor Comments (optional):

Reviewers' comments:

Reviewer's Responses to Questions

**Comments to the Author**

1. If the authors have adequately addressed your comments raised in a previous round of review and you feel that this manuscript is now acceptable for publication, you may indicate that here to bypass the “Comments to the Author” section, enter your conflict of interest statement in the “Confidential to Editor” section, and submit your "Accept" recommendation.

Reviewer #2: All comments have been addressed

2. Is the manuscript technically sound, and do the data support the conclusions?

Reviewer #2: (No Response)

3. Has the statistical analysis been performed appropriately and rigorously? 

Reviewer #2: (No Response)

4. Have the authors made all data underlying the findings in their manuscript fully available?

Reviewer #2: (No Response)

5. Is the manuscript presented in an intelligible fashion and written in standard English?

Reviewer #2: (No Response)

6. Review Comments to the Author

Reviewer #2: (No Response)

7. PLOS authors have the option to publish the peer review history of their article (what does this mean?). If published, this will include your full peer review and any attached files.

Reviewer #2: No

---

## [Editor Report · Acceptance letter]

5 May 2023

PONE-D-22-31656R1 

A-B-3 – associations and dissociations of reading and arithmetic:
Is domain-specific prediction outdated? 

Dear Dr. Jöbstl:

I'm pleased to inform you that your manuscript has been deemed suitable for publication in PLOS ONE. Congratulations! Your manuscript is now with our production department. 

Kind regards, 

on behalf of

Dr. Madelon van den Boer 

Academic Editor

PLOS ONE